# Temporal trends in African healthcare system capacities in prevention, detection, response, and sustainability between 2010 and 2023

Pratik Sharma[1], Supriya D. Mahajan[2], Ravikumar Aalinkeel[2,3,4,5]*

1 Gifted Math Program, Graduate School of Education, University at Buffalo, Buffalo, New York, United States of America, 2 Department of Medicine, University at Buffalo, Buffalo, New York, United States of America, 3 Department of Urology, Jacobs School of Medicine and Biomedical Sciences, University at Buffalo, Buffalo, New York, United States of America, 4 Department of Medicine, Division of Allergy, Immunology, and Rheumatology, Jacobs School of Medicine and Biomedical Sciences, University at Buffalo, Buffalo, New York, United States of America, 5 Department of Medicine, VA Western New York Healthcare System, Buffalo, New York, United States of America

* ra5@buffalo.edu

## Abstract

This analysis evaluated self-reported national capacities for prevention, detection, response, and sustainability in 54 African countries from 2013 to 2023 using data from WHO State Party Annual Report (SPAR) submissions. Countries were grouped into five UN-defined regions. Global averages were used as external benchmarks. Four capacity domains were scored using SPAR indicators. Regional and temporal differences were analyzed using ANOVA with Tukey's multiple comparisons. Time trends from 2010 to 2023 were assessed using linear regression. Pre- and post-COVID-19 changes were compared using 2018 and 2023 data. Global scores were consistently higher than African regional means across all capacities. In 2023, African preventive capacity was 63.88 globally and 52.66 (p < 0.0001). Northern Africa scored highest within Africa (67.22), while Middle Sub-Saharan Africa remained lowest (34.07; p = 0.0134 vs Northern). Detection capacity in Africa rose from 56.67 (2013) to 63.12 (2023), still trailing the global score of 72.21 (p = 0.0001). Response capacity increased sharply in Africa in the post-pandemic period, from 38.48 in 2018 to 59.86 in 2023 (p = 0.0014), led by Northern Africa (73.75). Sustainability scores improved modestly in Africa (46.28 in 2018 to 54.78 in 2023; p = 0.0006), with Middle Sub-Saharan Africa lowest at 43.81. Full-period regression analysis revealed significant negative trends in prevention capacity for Africa overall and several subregions and positive trends in detection capacity for Northern, Middle, and Western Africa. Sensitivity analyses restricted to 2018–2023 revealed prevention trends reversed to positive; response and sustainability gains were concentrated in the post-pandemic period. Between 2013 and 2023, African regions showed incremental gains across all capacity domains, with the largest post-pandemic improvement in response capacity. However, persistent performance gaps remain, especially in Middle Sub-Saharan

**Data availability statement:** All data underlying the results reported in this manuscript are publicly available from the World Health Organization (WHO) State Party Annual Report (SPAR) database at https://extranet.who.int/sph/spar. Country-level scores were downloaded and processed to generate regional mean scores, perform ANOVA with Tukey's multiple comparisons, and fit linear regression models, as described in the Methods. The processed dataset supporting the findings of this study has been uploaded as Supporting Data.

**Funding:** The author(s) received no specific funding for this work.

**Competing interests:** The authors have declared that no competing interests exist.

Africa. The data support targeted investment in infrastructure, workforce, and diagnostics to close gaps and stabilize long-term health system performance.

## Introduction

The past decade has shifted the focus of African healthcare systems towards prevention, detection, response, and sustainability of health intervention. These factors have increased interest considering both endemic emergencies like the COVID-19 pandemic and the mpox virus along with other health challenges [1,2]. While global health initiatives have aimed to strengthen healthcare systems, significant disparities remain, particularly in low- and middle-income regions such as the various subsections of Sub-Saharan Africa. These disparities affect public health outcomes and the economic stability of these regions.

The World Health Organization (WHO) obtains vital information from states through State Party Annual Report (SPAR) submissions, which indicate how countries perceive their capacities to manage public health risks and emergencies [3,4]. The degree of progress in healthcare capacity development across African countries over the past decade, and the variation in that capacity between regions, has not been systematically quantified. This information gap limits regional and national planning for health emergency preparedness and response, particularly given the known impact of such events on health systems and economic stability.

Although global comparisons provide context, this research focuses on challenges and progress within Africa's various regions, including Northern Africa and Eastern, Middle, Southern, and Western Sub-Saharan Africa [5].The healthcare challenges of these regions are influenced by economic development, political stability, and resource availability. Kandel et al. (2020) discussed the role of economic factors in influencing the resilience of healthcare systems during the COVID-19 pandemic, noting that regions with weaker economies faced greater difficulties in sustaining healthcare interventions during prolonged crises [6]. Gao et al. (2023) explored the role of political stability in explaining the cross-country variation of COVID-19 pandemic outcomes, finding that unstable political situations including social conflict, government instability, and political corruption negatively affect public health [7]. These coinciding factors create region-specific obstacles to strengthening health security across the African continent, demonstrating the need for contextually tailored strategies.

Africa's economic state also influences its response to health emergencies. Most African countries operate with severely restricted healthcare budgets that hinder their ability to make long-term investments in healthcare infrastructure, workforce training and medical supplies, all of which are integral to emergency preparedness [8]. The COVID-19 pandemic coincided with heightened scrutiny of African countries' healthcare systems, revealing existing weaknesses and drawing attention to areas requiring investment and improvement [9].

Several initiatives, including those led by WHO, have aimed to strengthen healthcare readiness and international collaboration in Africa. However, the extent of measurable improvement in healthcare capacities remains insufficiently quantified.

This study analyzes healthcare system capacities in prevention, detection, response, and sustainability across five African regions using data from 2013 to 2023. The objective is to quantify regional trends and differences in these four domains and identify areas of progress and persistent limitations. The findings are intended to help strategic planning and resource allocation for health system strengthening and emergency preparedness across the continent.

## Methods

**Study population and regional classification.** This study evaluated healthcare capacities in 54 African countries using data from the World Health Organization (WHO) State Party Annual Report (SPAR) submissions. Countries were grouped into five regions and were determined in accordance with United Nations classifications, as follows: Northern Africa is comprised of Algeria, Egypt, Libya, Morocco, Sudan, and Tunisia; Eastern Sub-Saharan Africa includes Burundi, Comoros, Djibouti, Eritrea, Ethiopia, Kenya, Madagascar, Malawi, Mauritius, Mozambique, Rwanda, Seychelles, South Sudan, Uganda, Tanzania, Zambia, and Zimbabwe; Middle Sub-Saharan Africa consists of Angola, Cameroon, Central African Republic, Chad, Congo, Democratic Republic of the Congo, Equatorial Guinea, Gabon, and São Tomé and Príncipe; Southern Sub-Saharan Africa is made up of Botswana, Eswatini, Lesotho, Namibia, and South Africa; and Western Sub-Saharan Africa encompasses Benin, Burkina Faso, Cabo Verde, Côte d'Ivoire, Gambia, Ghana, Guinea, Guinea-Bissau, Liberia, Mali, Mauritania, Niger, Nigeria, Senegal, Sierra Leone, and Togo [5]. A map comprised of these nations, excluding Seychelles and São Tomé and Príncipe, is displayed in Fig 1. Global data, comprising 195 countries including high-income nations, were used for benchmark comparisons (S1-S14 Data).

**Data source and capacity domains.** SPAR submissions provided country-level self-assessments of capacities relevant to the International Health Regulations (IHR). Four functional domains were assessed: capacity to prevent, detect, respond, and sustain. Although SPAR data were available from 2010 onward and are included in the regression trend

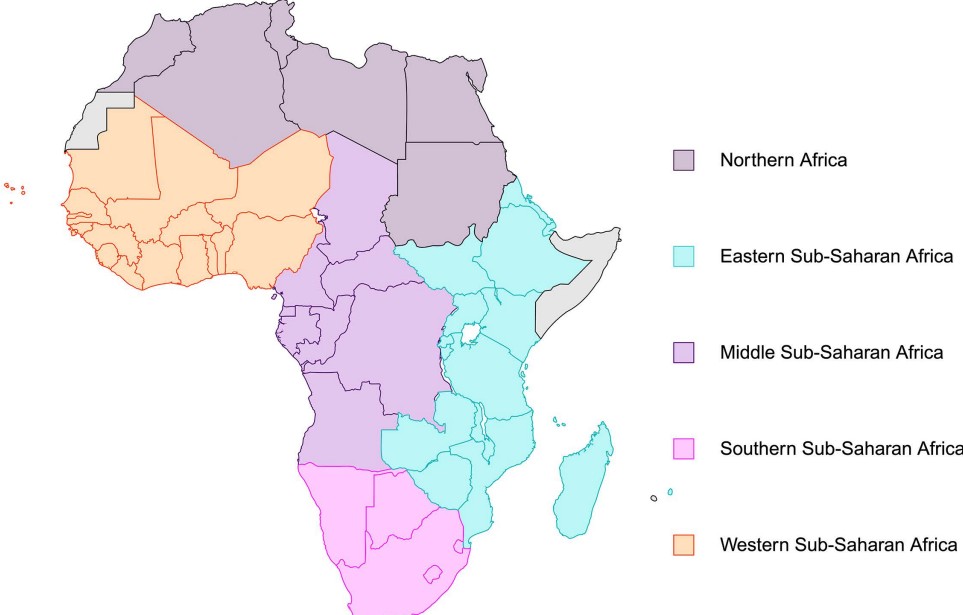

**Fig 1. Map displaying the nations, excluding Seychelles and São Tomé and Príncipe, which are not pictured, comprising the five United Nations-defined regions used for capacity score aggregation: Northern Africa (dark purple), Eastern Sub-Saharan Africa (light blue), Middle Sub-Saharan Africa (lavender), Southern Sub-Saharan Africa (pink), and Western Sub-Saharan Africa (orange).** All highlighted regions constitute the greater African region and are included in the Global group. Created in BioRender. Sharma, P. (2026) https://BioRender.com/4rxilwj.

analyses, cross-sectional comparative analyses and ANOVA-based regional comparisons use 2013 as the earliest time point, reflecting the first year with complete domain-level data across all five regions. The prevention domain included indicators related to risk communication, food safety, disease prevention, infection control, and measures associated with international travel. Detection was evaluated using metrics capturing disease surveillance systems, laboratory diagnostic capacity, early warning functions, and specimen transport systems. The response domain included indicators assessing emergency planning and response operations, resource logistics, community engagement, and health service readiness. Sustainability captures the health system's capacity to maintain essential public health functions over time, independent of acute emergency response. This domain reflects structural and enabling components such as health financing stability, workforce retention, service continuity, and supply chain resilience, emphasizing the ability to sustain preparedness gains beyond episodic shocks.

**Indicator Evolution and Comparability Over Time.** The assessment of healthcare capacities used data from WHO State Party Annual Report (SPAR) submissions, recognizing that indicator definitions and composition evolved across three reporting periods (2010–2017, 2018–2020, and 2021–2023). While the four functional domains—prevention, detection, response, and sustainability—remained conceptually stable, individual indicators within these domains were added, removed, or redefined over time (Table 1).

To address this evolution, harmonization was performed at the level of the four functional domains rather than at the individual indicator level. Indicators were mapped to domains based on their conceptual role within the International Health Regulations framework in effect at the time of reporting. Indicators were included only in the years in which they were defined and available, without retroactive rescaling or normalization across reporting periods.

For each country and year, domain scores were calculated as the mean of all indicators contributing to that domain in that reporting year. These country-level domain scores were then aggregated to derive regional mean values. This domain-level aggregation preserves internal consistency within each reporting year while accommodating documented changes in SPAR indicator composition over time. Accordingly, longitudinal trends are interpreted as changes in overall functional capacity rather than as changes in individual indicators. Of the three reporting transitions, the shift from the 2017–2018 framework involved the most substantial restructuring in indicator composition, including the addition, removal, and redefinition of multiple indicators across all four domains (Table 1). Observed score changes around this transition may therefore partly reflect measurement revision rather than true changes in underlying preparedness capacity. To address this, a sensitivity analysis was performed restricting regression analyses to the 2018–2023 period; results are described in the Results and S1 Table.

**Domain score construction and weighting considerations**. Domain scores were calculated using unweighted means of contributing indicators within each reporting year. Equal weighting was selected to avoid introducing subjective assumptions regarding the relative importance of individual indicators, particularly given periodic revisions in indicator definitions and scope within the SPAR framework. Because SPAR indicators are ordinal, self-reported, and designed to reflect complementary aspects of preparedness rather than hierarchical importance, unweighted averaging provides a transparent and reproducible summary measure of functional capacity.

**Score calculation and regional aggregation.** For each country, domain-specific scores were calculated by averaging the values of all corresponding indicators. These national averages were then aggregated to derive regional mean scores for each of the five African regions and for the global comparator group. This structure enabled comparisons across regions and over time, while maintaining consistency with the WHO evaluation framework. Regional standard deviations were calculated alongside means to characterize within-region variability in country-level scores and are presented in Fig 2 and S2 Table.

**Assessment of regional differences.** To evaluate differences in capacity scores among African regions, one-way Analysis of Variance (ANOVA) was conducted for the years 2013, 2018, and 2023 (S4, S9, S14 Data). When the ANOVA indicated significant variation between groups ($p < 0.05$), post hoc Tukey's Honest Significant Difference (HSD) tests

**Table 1. SPAR indicator composition by functional domain and reporting period (2010–2023).**

| 2010-2017 | | 2018-2020 | | 2021-2023 | |
|---|---|---|---|---|---|
| **Selected indicators** | | | | | |
| **Capacity to prevent** | | | | | |
| C.3.1.1 | Collaborative effort on activities to address zoonoses | C.3.1 | Collaborative effort on activities to address zoonoses | C.9.1 | IPC programmes |
| C.4.1.1 | Multisectoral collaboration mechanism for food safety events | C.4.1 | Multisectoral collaboration mechanism for food safety events | C.9.3 | Safe environment in health facilities |
| C.9.2.1 | Capacity for infection prevention and control and chemical and radiation decontamination | C.9.2 | Capacity for infection prevention and control and chemical and radiation decontamination | C.10.2 | Risk communication |
| C.10.1.1 | Capacity for emergency risk communications | C.10.1 | Capacity for emergency risk communications | C.11.1 | Core capacity requirements at all times for PoEs (airports, ports and ground crossings) |
| C.11.1.1 | Core capacity requirements at all times for designated airports, ports and ground crossings | C.11.1 | Core capacity requirements at all times for designated airports, ports and ground crossings | C.11.3 | Risk-based approach to international travel-related measures |
| | | | | C.13.1 | Multisectoral collaboration mechanism for food safety events |
| **Capacity to detect** | | | | | |
| C.5.1.1 | Specimen referral and transport system | C.5.1 | Specimen referral and transport system | C.4.1 | Specimen referral and transport system |
| C.5.3.1 | Access to laboratory testing capacity for priority diseases | C.5.3 | Access to laboratory testing capacity for priority diseases | C.4.3 | Laboratory quality system |
| C.6.1.1 | Early warning function: indicator-and event-based surveillance | C.6.1 | Early warning function: indicator-and event-based surveillance | C.4.4 | Laboratory testing capacity modalities |
| C.6.2.1 | Mechanism for event management (verification, risk assessment, analysis investigation) | C.6.2 | Mechanism for event management (verification, risk assessment, analysis investigation) | C.4.5 | Effective national diagnostic network |
| | | | | C.5.1 | Early warning surveillance function |
| | | | | C.5.2 | Event management (i.e., verification, investigation, analysis, and dissemination of information) |
| | | | | C.9.2 | Health care-associated infections (HCAI) surveillance |
| **Capacity to respond** | | | | | |
| C.8.1.1 | Planning for emergency preparedness and response mechanism | C.8.1 | Planning for emergency preparedness and response mechanism | C.7.1 | Planning for health emergencies |
| C.8.2.1 | Management of health emergency response operations | C.8.2 | Management of health emergency response operations | C.7.2 | Management of health emergency response |
| C.8.3.1 | Emergency resource mobilization | C.8.3 | Emergency resource mobilization | C.7.3 | Emergency logistic and supply chain management |
| C.9.1.1 | Case management capacity for IHR relevant hazards | C.9.1 | Case management capacity for IHR relevant hazards | C.8.1 | Case management |
| C.9.2.1 | Capacity for infection prevention and control and chemical and radiation decontamination | C.9.2 | Capacity for infection prevention and control and chemical and radiation decontamination | C.8.2 | Utilization of health services |

*(Continued)*

**Table 1.** (Continued)

| 2010-2017 | | 2018-2020 | | 2021-2023 | |
|---|---|---|---|---|---|
| **Selected indicators** | | | | | |
| **Capacity to prevent** | | | | | |
| C.11.2.1 | Effective public health response at points of entry | C.11.2 | Effective public health response at points of entry | C.10.3 | Community engagement |
| | | | | C.11.2 | Public health response at points of entry |
| | | | | C.12.1 | One Health collaborative efforts across sectors on activities to address zoonoses |
| **Capacity to sustain** | | | | | |
| C.1.3.1 | Financing mechanism and funds for timely response to public health emergencies | C.1.3 | Financing mechanism and funds for timely response to public health emergencies | C.1.1 | Policy, legal and normative instruments |
| C.2.1.2 | Multisectoral IHR coordination mechanisms | C.2.2 | Multisectoral IHR coordination mechanisms | C.2.2 | Multisectoral IHR coordination mechanisms |
| C.7.1.1 | Human resources for the implementation of IHR capacities | C.7.1 | Human resources for the implementation of IHR capacities | C.2.3 | Advocacy for IHR implementation |
| C.8.3.1 | Emergency resource mobilization | C.8.3 | Emergency resource mobilization | C.3.1 | Financing for IHR implementation |
| C.9.3.1 | Access to essential health services | C.9.3 | Access to essential health services | C.3.2 | Financing for Public Health Emergency Response |
| C.1.1.1 | Legislation, laws, regulations, administrative requirements, policies or other government instruments in place are sufficient for implementation of IHR | C.1.1 | Legislation, laws, regulations, policy, administrative requirements or other government instruments to implement the IHR | C.6.1 | Human resources for implementation of IHR |
| CC.1.2.1 | Financing for the implementation of IHR capacities | C.1.2 | Financing for the implementation of IHR capacities | C.8.3 | Continuity of essential health services (EHS) |

were used to identify statistically significant pairwise differences between regions for each domain. Prior to ANOVA, data distributions were assessed for approximate normality and homogeneity of variance using standard diagnostic checks; no substantial violations were identified that would preclude the use of parametric testing. No additional correction for multiple comparisons beyond Tukey's HSD was applied, as this procedure inherently controls the family-wise error rate across all pairwise regional comparisons.

**Pre-pandemic and post-pandemic comparison.** To evaluate the effects of the COVID-19 pandemic on healthcare system capacities, mean regional scores from 2018 and 2023 were compared (S9, S14 Data). In addition to absolute score differences, a five-point ordinal scale was applied to classify capacity levels across domains, with a score of 5 assigned to values ≥80, 4 for values ≥60, 3 for values ≥40, 2 for values ≥20, and 1 for values <20, consistent with ordinal capacity level frameworks used in WHO IHR monitoring [10]. This classification supported categorical interpretation of shifts in preparedness levels before and after the pandemic period. To provide further context to the observed post-pandemic changes in healthcare capacities, particularly in response, an exploratory analysis was conducted examining the relationship between regional improvements in response scores (2018–2023) and regional COVID-19 burden data. While comprehensive mortality and case count data across all 54 African countries and five regions over the entire pandemic period are complex to standardize, publicly available data from sources like the Johns Hopkins University Coronavirus Resource Center were utilized to approximate regional pandemic impact [11].

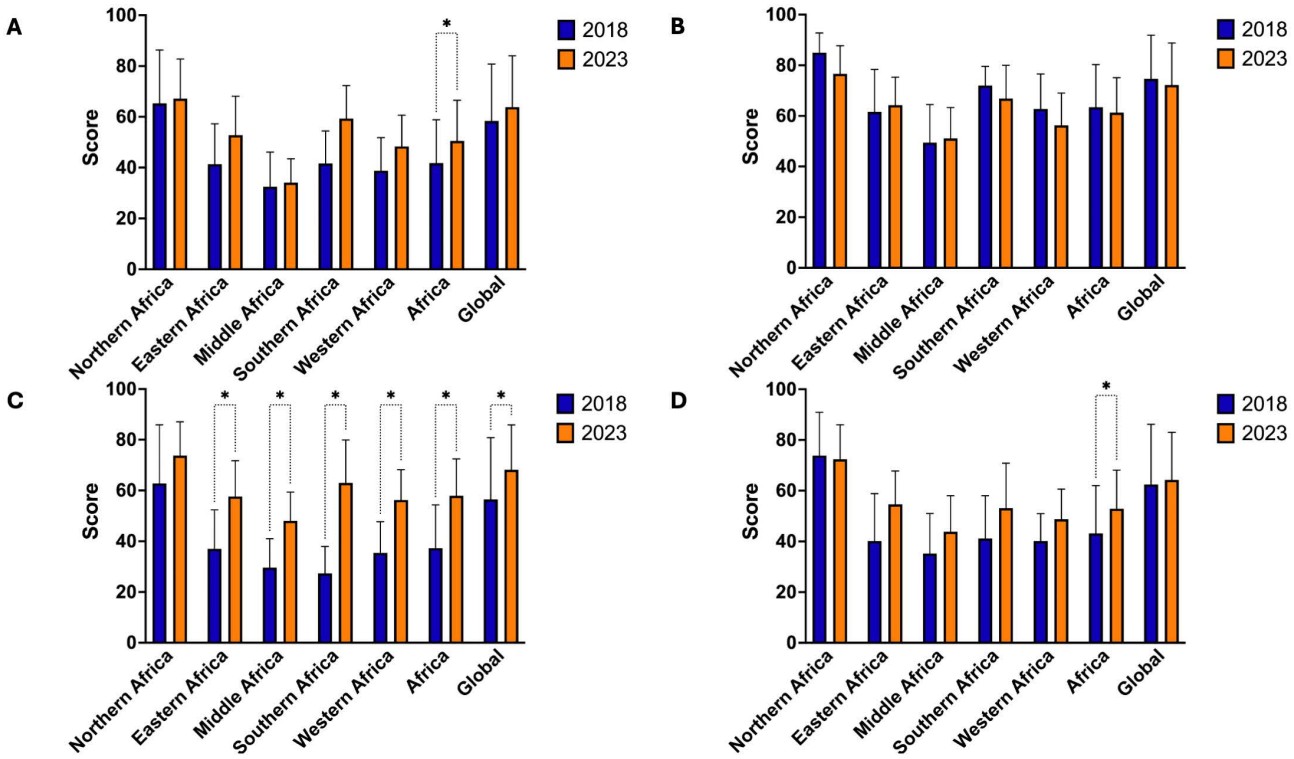

**Fig 2. Bar graphs comparing regional and global capacities in 2018 in 2023.** (A) represents prevention capacities; (B) depicts detection capacities; (C) presents response capacities; and (D) demonstrates sustainability capacities, evaluating long-term resilience in health systems. The data spans six geographic regions: Northern Africa, Eastern Africa, Middle Africa, Southern Africa, Western Africa, and the overall African average, alongside a global average. The y-axis indicates capacity scores ranging from 0 to 100, while the x-axis categorizes the regions. Blue bars correspond to scores in 2018, while orange bars represent scores in 2023, with error bars representing the standard deviation. Statistical significance was assessed using one-way ANOVA with Tukey's multiple comparisons test to account for pairwise differences across regions and years. Statistical significance is denoted by asterisk.

**Trend analysis using regression modeling.** To assess trends in healthcare system capacities over time, linear regression analyses were performed for each of the four domains using annual regional mean values from 2010 to 2023. The independent variable was year, and the dependent variable was the domain-specific score for each region. Regression models were fit using the Ordinary Least Squares (OLS) method. Model outputs included R-squared values, regression coefficients, and p-values, which were used to evaluate the strength and statistical significance of trends within each region and domain. Regional mean-level regression was selected because the primary unit of analysis in this study is the regional aggregate rather than the individual country. This approach provides a transparent and interpretable summary of continent-wide and regional-level preparedness trajectories consistent with the study's descriptive aims. It does not account for the panel structure of the underlying country-year data, serial correlation across time points, or within-region heterogeneity; a fixed or random effects panel model would be more appropriate for country-level inference, which was not the objective of this analysis.

As a sensitivity analysis to assess the potential impact of the 2018 SPAR framework transition on observed trends, linear regression was additionally performed restricting the analysis to the 2018–2023 period. This restricted analysis encompasses a single continuous reporting framework and avoids the structural discontinuity introduced at the 2017–2018 transition. Results from both the full-period and restricted analyses are presented in S1 Table and compared to assess directional consistency and robustness of the reported trends.

**Software and data processing.** All statistical analyses were performed using GraphPad Prism version 11 and Microsoft Excel. Data were manually reviewed and aligned prior to analysis to ensure consistency in regional grouping and indicator selection. Final scores were reported as regional means, with statistical comparisons based on significance thresholds of $p < 0.05$. The complete GraphPad Prism project file containing all analyses is provided as S15 Data. Original national datasets are additionally available as S1–S14 Data. Percent changes are reported descriptively alongside absolute score differences; 95% confidence intervals for pairwise mean differences from ANOVA are reported in S3-S5 Table, and regression slope confidence intervals are reported in Table 2 and S1 Table.

## Results

African regions differ substantially in baseline health system structure and economic capacity, with Northern Africa comprising predominantly upper-middle-income countries, while Middle, Eastern, and Western Sub-Saharan Africa include a higher proportion of low-income and lower-middle-income countries, providing important context for the observed regional differences in self-reported preparedness scores.

**Table 2. Linear regression estimates of temporal trends in SPAR functional domain scores by region, 2010–2023.**

| Domain | Region | β (slope) | 95% CI | R² | p-value |
|---|---|---|---|---|---|
| Prevent | Northern Africa | −0.214 | −1.386 to 0.957 | 0.013 | 0.697 |
| Prevent | Eastern Africa | −0.807 | −2.056 to 0.443 | 0.142 | 0.185 |
| Prevent | Middle Africa | −1.668 | −2.797 to −0.539 | 0.463 | 0.007 |
| Prevent | Southern Africa | −1.580 | −3.199 to 0.040 | 0.274 | 0.055 |
| Prevent | Western Africa | −1.274 | −2.459 to −0.090 | 0.314 | 0.037 |
| Prevent | Africa (overall) | −1.108 | −2.134 to −0.083 | 0.316 | 0.036 |
| Prevent | Global | −1.096 | −2.125 to −0.067 | 0.31 | 0.039 |
| Detect | Northern Africa | 1.608 | 0.549 to 2.667 | 0.477 | 0.006 |
| Detect | Eastern Africa | 1.377 | −0.263 to 3.016 | 0.218 | 0.092 |
| Detect | Middle Africa | 1.237 | 0.097 to 2.376 | 0.318 | 0.036 |
| Detect | Southern Africa | 1.005 | −0.606 to 2.616 | 0.133 | 0.199 |
| Detect | Western Africa | 1.625 | 0.070 to 3.181 | 0.302 | 0.042 |
| Detect | Africa (overall) | 1.37 | 0.439 to 2.302 | 0.461 | 0.008 |
| Detect | Global | 0.845 | −0.008 to 1.698 | 0.28 | 0.052 |
| Respond | Northern Africa | 0.553 | −0.656 to 1.763 | 0.076 | 0.339 |
| Respond | Eastern Africa | −0.257 | −1.609 to 1.095 | 0.014 | 0.686 |
| Respond | Middle Africa | −0.370 | −1.368 to 0.627 | 0.052 | 0.434 |
| Respond | Southern Africa | −1.412 | −3.649 to 0.825 | 0.136 | 0.194 |
| Respond | Western Africa | 0.185 | −1.055 to 1.425 | 0.009 | 0.751 |
| Respond | Africa (overall) | −0.260 | −1.460 to 0.940 | 0.018 | 0.645 |
| Respond | Global | 0.009 | −0.898 to 0.916 | <0.001 | 0.983 |
| Sustain | Northern Africa | 0.82 | −0.442 to 2.082 | 0.143 | 0.182 |
| Sustain | Eastern Africa | 0.702 | −0.800 to 2.205 | 0.08 | 0.329 |
| Sustain | Middle Africa | 0.368 | −0.403 to 1.140 | 0.083 | 0.319 |
| Sustain | Southern Africa | 0.228 | −1.901 to 2.356 | 0.005 | 0.82 |
| Sustain | Western Africa | 0.935 | −0.098 to 1.968 | 0.245 | 0.072 |
| Sustain | Africa (overall) | 0.611 | −0.359 to 1.580 | 0.136 | 0.195 |
| Sustain | Global | 0.369 | −0.248 to 0.985 | 0.124 | 0.217 |

**Preventive capacity across regions and over time.** Preventive capacity scores varied between African regions and the global benchmark (Table 3). In 2013, the Global mean was 77.37, compared to Africa's mean of 61.86 (p = 0.0004). Within Africa, Northern Africa had the highest preventive capacity (71.13), led by Morocco, and Middle Sub-Saharan Africa, with the worst result reported by Equatorial Guinea, had the lowest (45.25), though this difference was not statistically significant (p = 0.1881). By 2018, global preventive capacity declined to 58.38 by 24.5% and African to 44.05 by 28.9% (p < 0.0001). Northern Africa remained highest at 65.33 with an 8.2% decrease, with Algeria now leading the region, while Middle Sub-Saharan Africa, now with Central African Republic disclosing the worst score, declined to 32.44, decreasing by 28.3%, a significant difference (p = 0.0389). In 2023, global and African preventive scores improved to 63.88 and 52.66 with increases of 9.4% and 19.5%, respectively (p < 0.0001). Northern Africa, led by Egypt, reached 67.22, increasing by 2.9%, and Middle Sub-Saharan Africa, with Central African Republic remaining the lowest score, improved to 34.07 by 5.0%, with their inter-regional difference remaining significant (p = 0.0134).

**Detection capacity trends and regional variation.** Detection capacity exhibited regional variation (Table 3). In 2013, the Global mean was 71.98 compared to Africa's 56.67 (p = 0.0212). Within Africa, Northern Africa registered 67.33, led by Egypt and Morocco, and Middle Sub-Saharan Africa, with Equatorial Guinea and Sao Tome and Principe reporting the lowest scores, recorded 46.13, with no statistically significant difference (p = 0.7370). By 2018, global detection capacity increased to 74.67 by 3.7% and African to 66.44 by 17.2% (p = 0.0004). Northern Africa, now with Egypt leading, reached 85 with an increase of 26.2%, while Middle Sub-Saharan Africa, with Central African Republic joining the other two nations as the least scores in the region, was lowest at 49.44, increasing by 7.2%, a significant difference (p = 0.0013). In 2023, the Global score declined to 72.21 by 3.3%, and Africa's to 63.12 by 5.0% (p = 0.0001). Northern Africa recorded 76.67, decreasing by 9.8%, led by Tunisia, and Middle Sub-Saharan Africa, now with Chad reporting the lowest score, improved to 51.11 by 3.4%. The disparity between the highest and lowest regions in Africa remained statistically significant (p = 0.0304).

**Response capacity: regional trends and post-pandemic changes.** Response capacity scores varied regionally (Table 3). In 2013, the Global mean was 69.62, compared to Africa's 51.91 (p = 0.001). Western Sub-Saharan Africa, led by Côte d'Ivoire, registered 60.34, and Middle Sub-Saharan Africa, with Sao Tome and Principe reporting the lowest score, recorded 47.22, with no statistically significant difference (p = 0.9162). In 2018, Global scores declined to 56.48 by 18.9% and African to 38.48 by 25.9% (p < 0.0001). Northern Africa was highest at 62.78, led by Egypt, and Southern Sub-Saharan Africa, with the worst score from Lesotho, recorded 27.33 (p = 0.1023). By 2023, global response capacity rose to 68.13 by 20.6%, and Africa's mean improved to 59.86 by 55.6% (p = 0.0014), coinciding with the post-COVID-19 pandemic period during which many countries substantially scaled up emergency preparedness investments. Northern Africa led with 73.75, increasing by 17.5%, still led by Egypt, while Middle Sub-Saharan Africa, with the lowest scores from Chad and Democratic Republic of the Congo, had the lowest score at 51.11 (p = 0.0516).

**Sustainability capacity and long-term readiness.** Sustainability scores in African regions were lower than the global mean (Table 3). In 2013, the Global score was 64.76, compared to Africa's 45.65 (p = 0.0004). Within Africa, Northern Africa, led by Morocco, scored 68.38, and Middle Sub-Saharan Africa, with the worst score from Central African Republic, scored 31.97, with no significant difference. In 2018, the Global mean decreased to 62.48 by 3.5% and Africa's increased to 46.28 by 1.3% (p < 0.0001). Northern Africa improved to 73.81 by 7.9%, with Algeria now leading, while Middle Sub-Saharan Africa reached 35.24 by 10.2% (p = 0.0143); Central African Republic remained the worst score in this region. In 2023, the Global score increased to 64.24 by 2.8% and Africa's to 54.78 by 18.4% (p = 0.0006). Northern Africa scored 72.38, decreasing by 1.9%, with Egypt reporting the highest score, and Middle Sub-Saharan Africa, now with Chad and Congo joining Central African Republic with the lowest reported score, improved to 43.81 by 24.3%. This difference was statistically significant (p = 0.0320).

**Regional progress and stagnation across the decade.** From 2013 to 2023, the Global group showed fluctuation in capacity scores (S4, S9, S14 Data), particularly a drop in 2018 followed by recovery in 2023, most notably in response

**Table 3. Scores at global and regional levels from 2010 to 2023 in all capacities, with n representing sample size per region and year.**

| Year | Capacity to | Northern Africa | | Eastern Sub-Saharan Africa | | Middle Sub-Saharan Africa | | Southern Sub-Saharan Africa | | Western Sub-Saharan Africa | | AFRICA | | GLOBAL | |
|---|---|---|---|---|---|---|---|---|---|---|---|---|---|---|---|
| | | Mean | SD | Mean | SD | Mean | SD | Mean | SD | Mean | SD | Mean | SD | Mean | SD |
| 2010 | prevent | 72.16 | 25.3 | 52.1 | 33.96 | 45.8 | 26.84 | 68 | 23.62 | 62.7 | 31.09 | 60.15 | 28.16 | 68.57 | 29.23 |
| | detect | 64.5 | 30.69 | 30.42 | 31.22 | 23.75 | 23.11 | 40 | 34.12 | 37.19 | 36.33 | 39.17 | 31.09 | 55.89 | 34.56 |
| | respond | 65.1 | 32.08 | 52.54 | 35.87 | 41.75 | 34.25 | 61.44 | 27.73 | 48.47 | 31.52 | 53.86 | 32.29 | 59.43 | 33.1 |
| | sustain | 69.85 | 23.07 | 38.96 | 28.01 | 38 | 34.44 | 51.06 | 34.03 | 38.47 | 35.59 | 47.27 | 31.03 | 59.95 | 34.92 |
| 2011 | prevent | 55.23 | 38.25 | 63.15 | 29.07 | 51.18 | 29.79 | 76.67 | 22.36 | 53.97 | 34.09 | 60.04 | 30.71 | 71.44 | 30.19 |
| | detect | 52.33 | 34.93 | 56.13 | 28.74 | 35.83 | 32.83 | 64.17 | 20.17 | 38.21 | 25.86 | 49.33 | 28.5 | 64.99 | 31.01 |
| | respond | 65.25 | 36.77 | 61.92 | 36.35 | 46.17 | 35.84 | 72.92 | 27.1 | 49.23 | 32.61 | 59.1 | 33.73 | 63.72 | 32.95 |
| | sustain | 48.08 | 28.1 | 41.13 | 36.89 | 33.97 | 36.72 | 41.25 | 39.26 | 33.13 | 32.3 | 39.51 | 34.65 | 54.84 | 38.36 |
| 2012 | prevent | 63 | 32.32 | 63.67 | 29.28 | 62.85 | 29.6 | 60.1 | 34.43 | 52.7 | 33.85 | 60.46 | 31.9 | 75.85 | 29.9 |
| | detect | 55.5 | 33.41 | 61.83 | 33.34 | 58.88 | 37.99 | 79.25 | 17.58 | 32.67 | 24.46 | 57.63 | 29.35 | 68.74 | 31.4 |
| | respond | 52.05 | 41.39 | 56.92 | 34.94 | 52.38 | 31.13 | 56 | 41.18 | 49.67 | 34.02 | 53.4 | 36.53 | 69.13 | 32.69 |
| | sustain | 57.58 | 29.07 | 46 | 37.11 | 48 | 39.12 | 29 | 32.92 | 23.63 | 29.04 | 40.84 | 33.45 | 60.74 | 37.66 |
| 2013 | prevent | 71.13 | 28.42 | 62.83 | 35.36 | 45.25 | 32.16 | 65.75 | 32.44 | 71 | 31.19 | 63.19 | 31.91 | 77.37 | 30.5 |
| | detect | 67.33 | 34.74 | 56.29 | 30.06 | 46.13 | 34.67 | 50.13 | 39.84 | 67.31 | 27.51 | 57.44 | 33.36 | 71.98 | 30.44 |
| | respond | 58.25 | 38.14 | 49.75 | 37.56 | 47.22 | 34.65 | 58.75 | 35.71 | 60.34 | 37.58 | 54.86 | 36.73 | 69.62 | 34.19 |
| | sustain | 68.38 | 28.22 | 45.16 | 36.44 | 31.97 | 33.43 | 40.56 | 33.64 | 50.78 | 37 | 47.37 | 33.74 | 64.76 | 36.56 |
| 2014 | prevent | 86.47 | 21.32 | 66.85 | 37.42 | 46.53 | 34.28 | 78.27 | 31.86 | 54.26 | 35.29 | 65.67 | 32.03 | 79.7 | 29.41 |
| | detect | 78.75 | 33.91 | 69.81 | 29.01 | 48.5 | 40.54 | 81.67 | 22.9 | 54.1 | 29.11 | 66.56 | 31.09 | 76.15 | 27.8 |
| | respond | 79.4 | 21.44 | 57.5 | 37.51 | 42.54 | 38.73 | 78.17 | 33.6 | 49.88 | 36.26 | 61.5 | 33.51 | 73.12 | 32.14 |
| | sustain | 78.08 | 20.75 | 56.56 | 38.89 | 39.08 | 36.86 | 70.42 | 32.78 | 44.9 | 33.89 | 57.81 | 32.64 | 68.87 | 34.64 |
| 2015 | prevent | 73.2 | 12.09 | 68.09 | 36.76 | 56.07 | 32.43 | 70 | 35.32 | 58.17 | 40.59 | 65.1 | 31.44 | 81.85 | 28.68 |
| | detect | 68 | 17.45 | 79.72 | 19.57 | 53.5 | 39.11 | 58.83 | 39.52 | 61.92 | 27.34 | 64.39 | 28.6 | 81.89 | 23.84 |
| | respond | 67.13 | 15.69 | 65.03 | 33.31 | 51 | 33.97 | 74.75 | 38.34 | 45.88 | 38.47 | 60.76 | 31.96 | 75.48 | 30.99 |
| | sustain | 81.63 | 16.75 | 68.17 | 34.12 | 32.17 | 35.37 | 67.25 | 39.33 | 51.08 | 42.71 | 60.06 | 33.66 | 71.70 | 34.23 |
| 2016 | prevent | 84.08 | 28.02 | 77.4 | 36.17 | 55.77 | 35.59 | 81.7 | 30.27 | 63.43 | 37.19 | 72.48 | 33.45 | 82.47 | 28.49 |
| | detect | 65.3 | 37.93 | 83.67 | 23.95 | 48.79 | 28.84 | 77.75 | 18.25 | 76.06 | 26.04 | 70.31 | 27 | 81.29 | 24.75 |
| | respond | 78.81 | 25.34 | 74.25 | 30.19 | 45.07 | 33.29 | 79.88 | 34.22 | 52.94 | 37.37 | 66.19 | 32.08 | 75.55 | 31.25 |
| | sustain | 75.9 | 22.59 | 76.33 | 32.07 | 34.93 | 31.51 | 81.63 | 28.75 | 48.66 | 40.35 | 63.49 | 31.06 | 70.73 | 35.27 |
| 2017 | prevent | 71.37 | 23.6 | 64.19 | 34.52 | 51.31 | 35.01 | 76.92 | 30.73 | 64.06 | 34.32 | 65.57 | 31.64 | 78.14 | 30.49 |
| | detect | 76.67 | 27.58 | 64.21 | 30.04 | 47.11 | 32.43 | 70.9 | 25.3 | 62.75 | 30.55 | 64.33 | 29.18 | 74.06 | 29.01 |
| | respond | 80 | 28.71 | 58.87 | 34.36 | 45.22 | 34.53 | 70.95 | 34.59 | 54.56 | 38.43 | 61.92 | 34.12 | 71.98 | 32.61 |
| | sustain | 82.46 | 18.65 | 49.24 | 37.82 | 35.69 | 33.19 | 63.3 | 40.74 | 48.98 | 37.56 | 55.93 | 33.59 | 65.77 | 36.26 |
| 2018 | prevent | 65.33 | 24.03 | 42.12 | 23.51 | 32.44 | 23.85 | 41.6 | 22.3 | 38.75 | 21.43 | 44.05 | 23.02 | 58.38 | 28.96 |
| | detect | 85 | 20.3 | 62.94 | 24.5 | 49.44 | 26.4 | 72 | 20.93 | 62.81 | 21.93 | 66.44 | 22.81 | 74.67 | 24.9 |
| | respond | 62.78 | 28.96 | 37.25 | 21.62 | 29.63 | 19.71 | 27.33 | 20.67 | 35.42 | 20.21 | 38.48 | 22.23 | 56.48 | 30.11 |
| | sustain | 73.81 | 24.76 | 41.01 | 26.28 | 35.24 | 24.02 | 41.14 | 26.98 | 40.18 | 20.88 | 46.28 | 24.59 | 62.48 | 30.45 |
| 2019 | prevent | 70 | 22.85 | 49.88 | 21.96 | 29.33 | 18.39 | 43.2 | 23.58 | 39.5 | 19.09 | 46.38 | 21.17 | 62.03 | 27.74 |
| | detect | 80 | 23.57 | 67.06 | 23.38 | 46.67 | 23.9 | 69 | 24.69 | 62.5 | 20.93 | 65.05 | 23.29 | 75.89 | 23.71 |
| | respond | 60 | 29.17 | 43.33 | 24.06 | 29.26 | 16.81 | 34.67 | 25.15 | 36.25 | 16.5 | 40.7 | 22.34 | 60.11 | 28.98 |
| | sustain | 75.71 | 22.27 | 48.07 | 27.47 | 30.79 | 21.2 | 43.43 | 28.07 | 40.36 | 20.09 | 47.67 | 23.82 | 65.52 | 28.85 |

*(Continued)*

**Table 3.** (Continued)

| Year | Capacity to | Northern Africa | | Eastern Sub-Saharan Africa | | Middle Sub-Saharan Africa | | Southern Sub-Saharan Africa | | Western Sub-Saharan Africa | | AFRICA | | GLOBAL | |
|---|---|---|---|---|---|---|---|---|---|---|---|---|---|---|---|
| | | Mean | SD | Mean | SD | Mean | SD | Mean | SD | Mean | SD | Mean | SD | Mean | SD |
| 2020 | prevent | 68.67 | 24.68 | 55.29 | 21.3 | 35.11 | 20.07 | 55.2 | 21.82 | 44.5 | 22.04 | 51.75 | 21.98 | 63.03 | 26.71 |
| | detect | 82.5 | 19.32 | 67.65 | 22.93 | 58.33 | 24.55 | 77 | 17.5 | 62.19 | 24.46 | 69.53 | 21.75 | 78.33 | 21.58 |
| | respond | 63.33 | 27.07 | 50.78 | 24.32 | 40.74 | 18.21 | 43.33 | 17.49 | 40.83 | 18.79 | 47.81 | 21.17 | 62.84 | 27.42 |
| | sustain | 71.9 | 19.02 | 52.27 | 25.66 | 40.95 | 20.77 | 54.29 | 26.38 | 42.14 | 19.97 | 52.31 | 22.36 | 66.24 | 27.37 |
| 2021 | prevent | 58.89 | 25.45 | 54.71 | 21.05 | 36.3 | 19.06 | 57.33 | 27.16 | 50.63 | 19.46 | 51.57 | 22.44 | 63.9 | 26.42 |
| | detect | 72.38 | 21.77 | 66.72 | 19.18 | 48.89 | 20.88 | 66.29 | 23.15 | 61.61 | 22.32 | 63.18 | 21.46 | 73.4 | 22.79 |
| | respond | 68.33 | 22.1 | 58.24 | 24.03 | 42.78 | 20.5 | 50.5 | 27.12 | 58.75 | 18.47 | 55.72 | 22.44 | 68.37 | 26.21 |
| | sustain | 69.05 | 24.49 | 51.93 | 19.01 | 40 | 16.85 | 45.71 | 26.82 | 48.93 | 18.95 | 51.12 | 21.22 | 64.04 | 25.25 |
| 2022 | prevent | 66.11 | 21.26 | 55.29 | 22.19 | 38.89 | 18.8 | 54.67 | 24.6 | 48.75 | 21.09 | 52.74 | 21.59 | 65.38 | 25.85 |
| | detect | 74.76 | 22.52 | 68.91 | 18.54 | 57.14 | 19.63 | 70.29 | 20.22 | 59.64 | 20.79 | 66.15 | 20.34 | 75.56 | 21.47 |
| | respond | 71.25 | 19.69 | 58.24 | 24.03 | 47.22 | 22.09 | 60.5 | 25.26 | 61.56 | 19.85 | 59.75 | 22.18 | 70.17 | 25.37 |
| | sustain | 70 | 22.91 | 54.62 | 21.58 | 46.03 | 19.55 | 46.86 | 25.18 | 49.46 | 19.98 | 53.4 | 21.64 | 65.87 | 24.64 |
| 2023 | prevent | 67.22 | 22.88 | 54.31 | 21.13 | 34.07 | 15.84 | 59.33 | 23.77 | 48.33 | 20.45 | 52.66 | 20.82 | 63.88 | 25.4 |
| | detect | 76.67 | 20.13 | 64.71 | 19.78 | 51.11 | 22.94 | 66.86 | 25.64 | 56.25 | 21.23 | 63.12 | 21.94 | 72.21 | 23.13 |
| | respond | 73.75 | 20.39 | 58.24 | 24.03 | 48.06 | 22.18 | 63 | 26.01 | 56.25 | 20.45 | 59.86 | 22.61 | 68.13 | 25.03 |
| | sustain | 72.38 | 21.7 | 55.8 | 21.77 | 43.81 | 20.27 | 53.14 | 26.1 | 48.75 | 19.32 | 54.78 | 21.83 | 64.24 | 24.35 |

capacity (56.48 to 68.13, an increase of 20.6%). Africa followed a similar trend, but progress varied widely by region. Middle Sub-Saharan Africa showed minimal gains, with preventive capacity increasing only from 32.44 to 34.07 between 2018 and 2023, only a 5.0% gain. Northern Africa consistently scored highest across all domains. The data illustrate substantial intra-continental variation, particularly between Northern and Middle Sub-Saharan Africa. Within-region variability was also considerable, particularly in Eastern and Western Sub-Saharan Africa, where standard deviations around regional means were largest, indicating that aggregate scores mask meaningful heterogeneity in individual country performance within these regions (Fig 2, S2 Table).

**Capacity level changes in the post-pandemic period (2018–2023).** Comparative data between 2018 and 2023 (Fig 2) show changes in several domains (S9, S14 Data). Africa's response capacity increased from 38.48 to 59.86 by 55.6%, while Middle Sub-Saharan Africa improved from 29.63 to 48.06 by 62.2% and Northern Africa from 62.78 to 73.75 by 17.5%. Sustainability improved from 46.28 to 54.78 by 18.4%, with Northern Africa showing the largest increase. Detection capacity declined slightly from 66.44 to 63.12 by 5.0%. Middle Sub-Saharan Africa continued to lag in overall progress.

**Trend analysis from 2010 to 2023** Linear regression analysis over the full 2010–2023 period (Fig 3) revealed significant negative trends in prevention capacity for Africa overall (slope=-1.11, p=0.036), Middle Sub-Saharan Africa (slope=-1.67, p=0.007), and Western Sub-Saharan Africa (slope=-1.27, p=0.037), as well as globally (slope=-1.10, p=0.039). Detection capacity showed statistically significant positive trends in Northern Africa (slope=1.61, p=0.006), Middle Sub-Saharan Africa (slope=1.24, p=0.036), Western Sub-Saharan Africa (slope=1.63, p=0.042), and Africa overall (slope=1.37, p=0.008). Trends in response and sustainability were not statistically significant in any region over the full period. Regression coefficients with corresponding 95% confidence intervals for each regional model are provided in Table 2.

Sensitivity analyses restricting the regression to the 2018–2023 period revealed substantially different patterns, highlighting the influence of the 2018 framework transition on full-period estimates (S1 Table). For prevention, directional

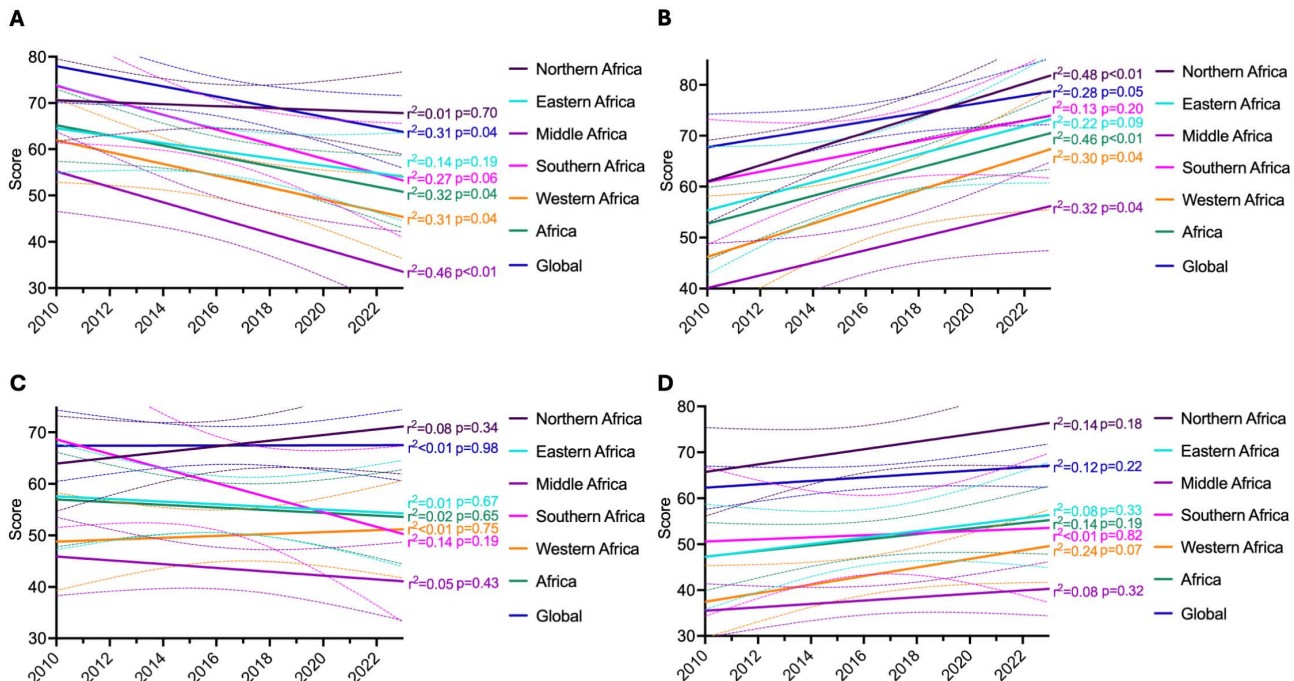

**Fig 3. Line graph displaying linear regression analysis of regional and global capacities to prevent health emergencies from 2010 to 2023.** (A) represents the analysis of prevention capacities; (B) focuses on detection capacities; (C) examines response capacities; and (D) analyzes sustainability capacities. The plot includes data from six geographic categories: Northern Africa, Eastern Africa, Middle Africa, Southern Africa, Western Africa, the overall African average, along with the global average. Each line represents a distinct region, with p-values indicated to reflect the significance of the observed trends. 95% CI bands are depicted with dotted lines in corresponding colors. Statistical significance was assessed using linear regression models, with p-values calculated for each regional trend to highlight areas of significant change over time.

trends reversed across all regions, with Africa overall (slope = +1.77, p = 0.016), Southern Sub-Saharan Africa (slope = +3.58, p = 0.019), Western Sub-Saharan Africa (slope = +2.34, p = 0.024), and globally (slope = +1.10, p = 0.032) showing significant positive trends in the post-2018 period. This reversal indicates that the negative full-period prevention trends were driven largely by the 2018 framework transition rather than a genuine long-term decline in preventive capacity.

For response capacity, no region showed a significant trend over the full period; however, restricting the analysis to 2018–2023 revealed significant positive trends in all regions and globally, with Africa overall showing a slope of +4.91 (p = 0.001) and Southern Sub-Saharan Africa the steepest improvement (slope = +7.51, p < 0.001). This indicates that response capacity gains are concentrated entirely in the post-pandemic period and were not detectable in the full-period analysis. Similarly, sustainability showed no significant trends over the full period but significant positive post-2018 trends in Africa overall (slope = +1.67, p = 0.005), Eastern (slope = +2.67, p = 0.009), Middle (slope = +2.50, p = 0.039), and Western Sub-Saharan Africa (slope = +2.20, p = 0.012).

Detection capacity showed an inverse pattern. Full-period trends were positive and significant for several regions, but post-2018 trends were uniformly non-significant or negative, with Western Sub-Saharan Africa showing a significant negative slope (-1.20, p = 0.017). This suggests that detection gains were concentrated in the pre-2018 period and have plateaued or modestly declined since.

**Post-pandemic regional directionality of change.** As shown in Table 4, regional analysis of directionality from 2018 to 2023 demonstrated substantial variation (S9, S14 Data). Southern Sub-Saharan Africa reported improvement in response

**Table 4. Percentage of countries at global and regional levels showing a decrease, no change, or increase in preparedness capacity levels between 2018 and 2023. Data are shown as the proportion (%) of countries in each category, with the total number of reporting countries listed in the rightmost column.**

| Capacity to | DECREASE IN LEVEL FROM 2018-2023 | | | | NO CHANGE IN LEVEL FROM 2018-2023 | | | | INCREASE IN LEVEL FROM 2018-2023 | | | | TOTAL COUNTRIES REPORTED |
|---|---|---|---|---|---|---|---|---|---|---|---|---|---|
| | pre-vent | detect | respond | sus-tain | pre-vent | detect | respond | sus-tain | pre-vent | detect | respond | sus-tain | |
| Northern Africa | 33 | 50 | 17 | 50 | 33 | 50 | 50 | 33 | 33 | 0 | 33 | 17 | 6 |
| Eastern Sub-Saharan Africa | 12 | 29 | 6 | 12 | 29 | 35 | 24 | 24 | 59 | 35 | 71 | 65 | 17 |
| Middle Sub-Saharan Africa | 11 | 33 | 0 | 11 | 44 | 11 | 44 | 44 | 44 | 56 | 56 | 44 | 9 |
| Southern Sub-Saharan Africa | 0 | 60 | 0 | 20 | 40 | 20 | 0 | 40 | 60 | 20 | 100 | 40 | 8 |
| Western Sub-Saharan Africa | 12 | 41 | 0 | 6 | 29 | 41 | 29 | 41 | 53 | 12 | 65 | 47 | 16 |
| AFRICA | 13 | 40 | 4 | 15 | 34 | 34 | 30 | 36 | 53 | 26 | 66 | 49 | 53 |
| GLOBAL | 13 | 32 | 11 | 23 | 52 | 53 | 45 | 48 | 35 | 15 | 43 | 29 | 196 |

capacity across 100% of its countries. Middle Sub-Saharan Africa showed no recorded declines but the highest rates of stagnation across all domains. Eastern and Western Sub-Saharan Africa displayed strong improvement across all four capacities, with over 70% of countries in Eastern Africa reporting increased scores in both response and sustainability. In contrast, Northern Africa had the highest percentage of countries showing declines in detection and sustainability (50% each). Across Africa, 53% of countries improved in prevention and 66% in response. However, sustainability remained vulnerable, with 36% reporting no change and 15% reporting declines. Compared to global patterns, African countries demonstrated a higher rate of improvement in response capacity (66% vs. 43%) but lower performance in detection and sustainability, highlighting ongoing system fragility in these categories.

## Discussion

This study provides an assessment and quantification of healthcare capacities across different African regions over a decade, with a focus on prevention, detection, response, and sustainability. The findings reveal gains in all four capacities across African regions between 2013 and 2023, most notably in response capacity. Sensitivity analyses restricting regression to the post-2018 period clarified and, in some cases, strengthened these patterns. Prevention trends, which appeared negative over the full 2010–2023 period, reversed to significantly positive when restricted to 2018–2023, suggesting the earlier negative trajectory was driven largely by the 2018 SPAR framework transition rather than genuine deterioration. Response and sustainability gains emerged as entirely post-2018 phenomena; neither domain showed significant trends over the full period, but both showed strong and consistent positive trajectories within the restricted analysis. Conversely, detection capacity, which appeared to improve over the full period, showed plateau or modest decline post-2018, indicating that earlier gains have not been sustained. However, significant disparities persist within Africa, particularly when comparing Northern Africa and Middle Sub-Saharan Africa, reflecting broader challenges in building resilient and effective healthcare systems [12], especially in the context of unexpected health emergencies such as the COVID-19 pandemic [13] or the growing occurrence of mpox virus [2,14]. It should be noted that without a formal interrupted time-series or segmented regression design, causal attribution of these improvements to the COVID-19 pandemic cannot be established from the current analysis. The observed gains in response capacity are temporally associated with the pandemic period and may reflect heightened emergency preparedness investments during this time, but other concurrent factors cannot be excluded.

Regional differences in preparedness capacity should be interpreted in the context of structural and economic differences across the African continent. Northern Africa, which consistently demonstrated the highest scores across domains,

includes countries with relatively higher per-capita income, more established health infrastructure, and longer-standing investments in surveillance and service delivery [15]. In contrast, Middle Sub-Saharan Africa, which persistently ranked lowest, comprises countries facing compounded constraints related to lower income levels, fragile health systems, workforce shortages, and dependence on external financing [16]. Eastern and Western Sub-Saharan Africa exhibited intermediate performance with greater variability, reflecting mixed economic profiles, uneven system strengthening efforts, and differing capacity to translate post-pandemic investments into sustained preparedness gains [17]. Regional averages for these subregions therefore represent central tendency rather than uniform performance, and individual country trajectories may diverge considerably from the aggregate trend. Country-level means and standard deviations for all regions and time points are provided in S2 Table.

As demonstrated in the results, the capacity to prevent health issues, as measured by ability of risk communication, food safety, disease prevention programs, safety of health facility environments, and travel-related measures, showed a consistent gap between African regions and the global benchmark. Northern Africa consistently demonstrated higher preventive capacity than the rest of Africa throughout the entirety of the 2013–2023 period, aligning with its relatively stronger healthcare infrastructure and economic resources [15]. In comparison, Middle Sub-Saharan Africa's consistently low scores highlight the region's ongoing struggle to implement effective preventive measures [18]. These findings indicate that preventive capacity remains unevenly distributed across Africa, with resource allocation and economic stability playing important roles [19].

Detection capacity, critical for early identification of health threats, also showed significant regional disparities. Northern Africa's higher detection capacity for the whole of the 10 year period can be attributed to better-developed disease surveillance systems and diagnostic capabilities [20]. Contrastingly, the lower detection capacities in regions like Eastern and Middle Sub-Saharan Africa raise concerns about these areas' ability to effectively monitor and respond to emerging health threats. This gap in detection capacity is particularly concerning given the region's vulnerability to outbreaks of infectious diseases [9,21]. A modest decline in detection capacity observed in some regions in the post-pandemic period suggests prolonged strain on surveillance systems and reinforces the need for sustained investment in these capabilities [22].

The ability to respond to health crises is a crucial component of healthcare system resilience. The substantial improvement in global response capacity in the post-pandemic period, particularly noteworthy with Southern Sub-Saharan Africa reporting 100% improvement across its countries, coincides with an escalated focus on strengthening emergency preparedness [23]. Within Africa, response capacity improved across regions, reflecting targeted investments in healthcare infrastructure and workforce training [24]. However, improvements in Middle Sub-Saharan Africa may still not be sufficient to overcome persistent structural barriers to effective emergency response.

Sustainability, the ability to maintain healthcare services over time, remains a critical challenge for African regions, with scores consistently lower than the global mean, as shown in Results. Northern Africa's higher sustainability capacity is reflective of better health financing mechanisms and more resilient healthcare systems [25]. In contrast, Middle Sub-Saharan Africa's low sustainability scores point to the fragility of its healthcare systems, which are often dependent on external funding and lack long-term planning for health system strengthening [26]. Although global sustainability capacity improved in the post-pandemic period, notably low scores in regions such as Middle Sub-Saharan Africa highlight ongoing weaknesses and the need for sustained investment in long-term health system strength [27,28].

The trends observed over the decade highlight both progress and ongoing challenges in African healthcare systems. While Northern and Southern Sub-Saharan Africa have shown significant improvements across various capacities, the stagnation in regions like Middle Sub-Saharan Africa calls for targeted interventions. The data indicate that economic stability, infrastructure development, and resource allocation are key determinants of healthcare capacity [29].

The pre- and post-pandemic comparison (Table 4) further illustrates the uneven distribution of capacity changes across regions. While the post-pandemic period coincided with improvements in some regions, particularly in response and sustainability capacities, it also revealed existing vulnerabilities, particularly in detection systems. The slight decline in

detection capacity in some African regions in the post-pandemic period suggests prolonged strain on surveillance systems, reinforcing the need for sustained investment in detection capabilities [30,31].

The pre- and post-pandemic comparison (Table 4) further illustrates the uneven distribution of capacity changes across regions. While the post-pandemic period coincided with improvements in some regions, particularly in response and sustainability capacities, it also revealed existing vulnerabilities, particularly in detection systems. The slight decline in detection capacity in some African regions in the post-pandemic period suggests prolonged strain on surveillance systems, reinforcing the need for sustained investment in detection capabilities [30,31].

Although linear regression was used to summarize long-term trends, preparedness directions across regions were not strictly linear over time. Several domains demonstrated inflection points, particularly around the COVID-19 period, consistent with episodic shocks rather than gradual change. More complex time-series approaches, such as non-linear or segmented models, were not applied, as the primary objective was to note overall directionality and relative differences across regions rather than to model short-term changes. Accordingly, regression results are interpreted as indicators of broad temporal patterns rather than precise predictive relationships. Moderate explanatory power and variable regional trajectories emphasize heterogeneity in preparedness evolution and caution against overinterpretation of linear trends, particularly in the presence of episodic shocks such as COVID-19 [32–36].

These findings suggest region-specific priorities for strengthening preparedness across Africa. In Middle Sub-Saharan Africa, persistently low scores across prevention, detection, and response highlight the need for foundational investments in surveillance infrastructure, laboratory capacity, and core public health functions [37,38]. In Western and Eastern Sub-Saharan Africa, where response and detection improved more rapidly following the COVID-19 period, policy efforts may be most impactful if focused on workforce retention, financing stability, and integration of emergency gains into routine health system operations [39]. In Northern Africa, where preparedness scores were consistently higher, priorities may center on sustaining investments, addressing remaining gaps in system resilience, and strengthening cross-border coordination [40,41]. Together, these findings indicate that preparedness strategies should be tailored to regional contexts rather than applied uniformly across the continent.

## Limitations and Data Considerations

This study analyzes African healthcare system capacities using self-assessed data from WHO State Party Annual Report (SPAR) submissions, and results should be interpreted as reflecting perceived rather than independently validated preparedness capacity. A known limitation is that self-reported data may introduce bias, potentially overestimating capacities. Studies comparing SPAR with other tools like the Joint External Evaluation (JEE) or Global Health Security (GHS) Index have shown discrepancies, with self-reported scores sometimes higher than external assessments [42,43]. This concern is particularly relevant for the post-2020 data, during which preparedness metrics gained considerable political visibility in the context of the COVID-19 pandemic. Upward reporting pressure in this period may have contributed to the observed gains in response and sustainability scores between 2018 and 2023, and findings from this period should be interpreted with this in mind. Triangulation with externally validated preparedness indices such as the JEE or Global Health Security Index was not performed and represents an important direction for future work. The use of unweighted domain averages may mask differences in the relative contribution of individual indicators; however, this approach prioritizes transparency and longitudinal comparability given the evolving structure of the SPAR framework. Similarly, the regression analyses were conducted on annual regional means rather than on country-year panel data. This approach was intentional given the study's regional-level descriptive aims, but it does not account for serial correlation across time points or within-region country-level variability. Findings from the regression analyses should therefore be interpreted as characterizing broad regional trajectories rather than precise estimates of trend magnitude.

A further methodological consideration is the structural evolution of the SPAR reporting framework across the study period. The transition from the 2010–2017 to the 2018–2020 indicator set involved substantial changes in the composition of all four functional domains, introducing a potential discontinuity that cannot be fully resolved through domain-level

harmonization alone. The observed declines in several capacity scores around 2018 may reflect, at least in part, changes in what was being measured rather than genuine deterioration in preparedness capacity. To address this, sensitivity analyses were performed restricting linear regression to the 2018–2023 period, encompassing a single continuous reporting framework. These analyses revealed directional reversals in prevention trends and unmasked significant positive trajectories in response and sustainability that were not apparent in the full-period models, confirming that the 2018 transition materially influenced full-period estimates. Formal structural break testing was not performed, and the sensitivity analyses presented here should be considered partial rather than definitive. Longitudinal findings should therefore be interpreted as reflecting broad directional patterns in self-reported preparedness capacity rather than precise point estimates of change.

Despite these limitations, SPAR data was specifically chosen for its unique suitability for longitudinal trend analysis across a decade. Unlike the JEE's external snapshots, SPAR offers a more continuous and consistent reporting mechanism over the entire 2013–2023 period. This sustained reporting enabled systematic assessment of regional variation and perceived evolution in healthcare capacity development, as well as internal country efforts to meet IHR requirements.

In conclusion, this study documents persistent regional variation in healthcare system capacities across Africa in the domains of prevention, detection, response, and sustainability from 2013 to 2023. While some regions demonstrated measurable improvement, others showed limited or inconsistent progress, highlighting capacity domains and geographic areas where additional investment may be warranted.

## Supporting information

**S1 Table. Sensitivity analysis: linear regression of domain-specific capacity scores restricted to the 2018–2023 period compared with full-period (2010–2023) estimates.**
(XLSX)

**S2 Table. Regional mean capacity scores (±SD) for prevention, detection, response, and sustainability across African regions and global comparator in 2013, 2018, and 2023.**
(XLSX)

**S3 Table. Pairwise comparisons of regional capacity scores (2013): mean differences, 95% confidence intervals, and adjusted p-values from one-way ANOVA with Tukey's HSD.**
(XLSX)

**S4 Table. Pairwise comparisons of regional capacity scores (2018): mean differences, 95% confidence intervals, and adjusted p-values from one-way ANOVA with Tukey's HSD.**
(XLSX)

**S5 Table. Pairwise comparisons of regional capacity scores (2023): mean differences, 95% confidence intervals, and adjusted p-values from one-way ANOVA with Tukey's HSD.**
(XLSX)

**S1 Data. Country-level SPAR domain scores for African countries, 2010.** Excel file containing country-level scores (0–100) for prevention, detection, response, and sustainability derived from WHO State Party Annual Report (SPAR) submissions for 2010.
(XLSX)

**S2 Data. Country-level SPAR domain scores for African countries, 2011.** Excel file containing country-level scores (0–100) for prevention, detection, response, and sustainability derived from WHO State Party Annual Report (SPAR) submissions for 2011.
(XLSX)

**S3 Data. Country-level SPAR domain scores for African countries, 2012.** Excel file containing country-level scores (0–100) for prevention, detection, response, and sustainability derived from WHO State Party Annual Report (SPAR) submissions for 2012.
(XLSX)

**S4 Data. Country-level SPAR domain scores for African countries, 2013.** Excel file containing country-level scores (0–100) for prevention, detection, response, and sustainability derived from WHO State Party Annual Report (SPAR) submissions for 2013.
(XLSX)

**S5 Data. Country-level SPAR domain scores for African countries, 2014.** Excel file containing country-level scores (0–100) for prevention, detection, response, and sustainability derived from WHO State Party Annual Report (SPAR) submissions for 2014.
(XLSX)

**S6 Data. Country-level SPAR domain scores for African countries, 2015.** Excel file containing country-level scores (0–100) for prevention, detection, response, and sustainability derived from WHO State Party Annual Report (SPAR) submissions for 2015.
(XLSX)

**S7 Data. Country-level SPAR domain scores for African countries, 2016.** Excel file containing country-level scores (0–100) for prevention, detection, response, and sustainability derived from WHO State Party Annual Report (SPAR) submissions for 2016.
(XLSX)

**S8 Data. Country-level SPAR domain scores for African countries, 2017.** Excel file containing country-level scores (0–100) for prevention, detection, response, and sustainability derived from WHO State Party Annual Report (SPAR) submissions for 2017.
(XLSX)

**S9 Data. Country-level SPAR domain scores for African countries, 2018.** Excel file containing country-level scores (0–100) for prevention, detection, response, and sustainability derived from WHO State Party Annual Report (SPAR) submissions for 2018.
(XLSX)

**S10 Data. Country-level SPAR domain scores for African countries, 2019.** Excel file containing country-level scores (0–100) for prevention, detection, response, and sustainability derived from WHO State Party Annual Report (SPAR) submissions for 2019.
(XLSX)

**S11 Data. Country-level SPAR domain scores for African countries, 2020.** Excel file containing country-level scores (0–100) for prevention, detection, response, and sustainability derived from WHO State Party Annual Report (SPAR) submissions for 2020.
(XLSX)

**S12 Data. Country-level SPAR domain scores for African countries, 2021.** Excel file containing country-level scores (0–100) for prevention, detection, response, and sustainability derived from WHO State Party Annual Report (SPAR) submissions for 2021.
(XLSX)

**S13 Data. Country-level SPAR domain scores for African countries, 2022.** Excel file containing country-level scores (0–100) for prevention, detection, response, and sustainability derived from WHO State Party Annual Report (SPAR) submissions for 2022.
(XLSX)

**S14 Data. Country-level SPAR domain scores for African countries, 2023.** Excel file containing country-level scores (0–100) for prevention, detection, response, and sustainability derived from WHO State Party Annual Report (SPAR) submissions for 2023.
(XLSX)

**S15 Data. GraphPad Prism project file (.prism) containing all primary analyses, including one-way ANOVA with Tukey's HSD for 2013, 2018, and 2023; full-period linear regression (2010–2023); and sensitivity analysis regression restricted to 2018–2023.** All regional domain score datasets are embedded within the file.
(ZIP)

## Author contributions

**Conceptualization:** Supriya D. Mahajan, Ravikumar Aalinkeel.

**Data curation:** Pratik Sharma.

**Formal analysis:** Pratik Sharma.

**Investigation:** Pratik Sharma.

**Methodology:** Pratik Sharma.

**Supervision:** Supriya D. Mahajan, Ravikumar Aalinkeel.

**Validation:** Supriya D. Mahajan, Ravikumar Aalinkeel.

**Writing – original draft:** Pratik Sharma.

**Writing – review & editing:** Pratik Sharma.

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
