## [Decision Letter · Decision Letter 0]

17 Nov 2025

PGPH-D-25-02770

Temporal trends in African healthcare system capacities in prevention, detection, response, and sustainability between 2010 and 2023

Dear Dr. Sharma,

Thank you for submitting your manuscript to PLOS Global Public Health. After careful consideration, we feel that it has merit but does not fully meet PLOS Global Public Health’s publication criteria as it currently stands. Therefore, we invite you to submit a revised version of the manuscript that addresses the points raised during the review process.

Please note that we have only been able to secure a single reviewer to assess your manuscript. We are issuing a decision on your manuscript at this point to prevent further delays in the evaluation of your manuscript. Please be aware that the editor who handles your revised manuscript might find it necessary to invite additional reviewers to assess this work once the revised manuscript is submitted. However, we will aim to proceed on the basis of this single review if possible.

Could you please carefully revise the manuscript to address all comments raised?

We look forward to receiving your revised manuscript.

Kind regards,

Helen Howard

Staff Editor

Journal Requirements:

1. In the online submission form, you indicated that “All data underlying the results reported in this manuscript are publicly available from the World Health Organization (WHO) State Party Annual Report (SPAR) database at https://extranet.who.int/sph/spar. Country-level scores were downloaded and processed to generate regional mean scores, perform ANOVA with Tukey’s multiple comparisons, and fit linear regression models, as described in the Methods. The processed dataset and analysis outputs supporting the findings of this study are available from the corresponding author upon reasonable request.”.

3. Uploaded as supplementary information.

2. Some material included in your submission may be copyrighted. According to PLOS’s copyright policy, authors who use figures or other material (e.g., graphics, clipart, maps) from another author or copyright holder must demonstrate or obtain permission to publish this material under the Creative Commons Attribution 4.0 International (CC BY 4.0) License used by PLOS journals. Please closely review the details of PLOS’s copyright requirements here: PLOS Licenses and Copyright. If you need to request permissions from a copyright holder, you may use PLOS's Copyright Content Permission form.

Potential Copyright Issues:

a. Figure 1:: please (a) provide a direct link to the base layer of the map (i.e., the country or region border shape) and ensure this is also included in the figure legend; and (b) provide a link to the terms of use / license information for the base layer image or shapefile. We cannot publish proprietary or copyrighted maps (e.g. Google Maps, Mapquest) and the terms of use for your map base layer must be compatible with our CC-BY 4.0 license.

Additional Editor Comments (if provided):

Reviewers' comments:

Reviewer's Responses to Questions

**Comments to the Author**

1. Does this manuscript meet PLOS Global Public Health’s publication criteria? Is the manuscript technically sound, and do the data support the conclusions? The manuscript must describe methodologically and ethically rigorous research with conclusions that are appropriately drawn based on the data presented.

Reviewer #1: Yes

2. Has the statistical analysis been performed appropriately and rigorously?

Reviewer #1: Yes

3. Have the authors made all data underlying the findings in their manuscript fully available (please refer to the Data Availability Statement at the start of the manuscript PDF file)?

Reviewer #1: Yes

4. Is the manuscript presented in an intelligible fashion and written in standard English?

Reviewer #1: Yes

5. Review Comments to the Author

Reviewer #1: This manuscript presents a comprehensive and much-needed quantitative assessment of African countries’ health system capacities in prevention, detection, response, and sustainability from 2010 to 2023 using WHO SPAR data. It offers an important contribution to regional preparedness literature, especially given the limited longitudinal analyses of SPAR data in Africa.

However, before publication, the manuscript would benefit from clarifications on methodology, data comparability, and statistical interpretation, as well as improvements in structure and conciseness to strengthen its rigor and policy relevance.

• Clarify how SPAR data from 2010–2023 were harmonized, given the well-known indicator evolution. The authors provide a thoughtful explanation of indicator evolution, but please explicitly state how differences in indicator definitions were adjusted or normalized before aggregating domain scores.

• With the indicator evolution, how did that impact the four functional domains? Instead of describing in text, consider a table that also would show when each indicator was added to reflect the evolution

• The choice of averaging domain indicators (rather than weighting by importance or completeness) may bias trends. Was any sensitivity analysis conducted that would strengthen the results?

• The use of ANOVA and Tukey HSD is appropriate, but please confirm assumptions (normality, homogeneity of variance) were met.

• Linear regression is reported, but results (R² = 0.31–0.46) indicate only moderate explanatory power. Add discussion on the potential non-linearity of trends or whether other time-series models were tested.

• Include confidence intervals for regression coefficients to align with BMC’s statistical reporting standards.

• Can you comment on the regional differences? Add more description of the continent and the region, ie. Income groups of the different countries?

• Several statements in the results (e.g., “Africa’s response capacity increased sharply after COVID-19”) would benefit from quantitative qualifiers (absolute or relative percentage change).

• A clearer explanation of what “sustainability” captures conceptually would help readers unfamiliar with WHO SPAR’s evolving framework.

The discussion appropriately references the contextual challenges of African health systems, but could be tightened to focus on the data’s empirical implications.

• Consider adding a short policy paragraph summarizing actionable regional priorities (e.g., laboratory networking in Middle Africa, workforce retention strategies in Western Africa).

• Tables 1 and 2 should include sample size (n countries) per region and year.

• Figures 2–3: Add error bars or 95% CI bands for clarity.

• Shorten the discussion to remove overlapping sentences and tone down the conclusion.

6. PLOS authors have the option to publish the peer review history of their article (what does this mean?). If published, this will include your full peer review and any attached files.

**Do you want your identity to be public for this peer review?** For information about this choice, including consent withdrawal, please see our Privacy Policy.

Reviewer #1: No

Figure Resubmissions:

---

## [Decision Letter · Decision Letter 1]

19 Feb 2026

PGPH-D-25-02770R1

Temporal trends in African healthcare system capacities in prevention, detection, response, and sustainability between 2010 and 2023

Dear Dr. Sharma,

Thank you for submitting your manuscript to PLOS Global Public Health. After careful consideration, we feel that it has merit but does not fully meet PLOS Global Public Health’s publication criteria as it currently stands. Therefore, we invite you to submit a revised version of the manuscript that addresses the points raised during the review process.

A letter that responds to each point raised by the editor and reviewer(s). You should upload this letter as a separate file labeled 'Response to Reviewers'.

We look forward to receiving your revised manuscript.

Kind regards,

Helen Howard

Staff Editor

Journal Requirements:

Additional Editor Comments (if provided):

Reviewers' comments:

Reviewer's Responses to Questions

**Comments to the Author**

1. If the authors have adequately addressed your comments raised in a previous round of review and you feel that this manuscript is now acceptable for publication, you may indicate that here to bypass the “Comments to the Author” section, enter your conflict of interest statement in the “Confidential to Editor” section, and submit your "Accept" recommendation.

Reviewer #1: All comments have been addressed

Reviewer #2: All comments have been addressed

2. Does this manuscript meet PLOS Global Public Health’s publication criteria? Is the manuscript technically sound, and do the data support the conclusions? The manuscript must describe methodologically and ethically rigorous research with conclusions that are appropriately drawn based on the data presented.

Reviewer #1: Yes

Reviewer #2: Yes

3. Has the statistical analysis been performed appropriately and rigorously?

Reviewer #1: Yes

Reviewer #2: No

4. Have the authors made all data underlying the findings in their manuscript fully available (please refer to the Data Availability Statement at the start of the manuscript PDF file)?

Reviewer #1: Yes

Reviewer #2: Yes

5. Is the manuscript presented in an intelligible fashion and written in standard English?

Reviewer #1: Yes

Reviewer #2: Yes

6. Review Comments to the Author

Reviewer #1: The manuscript has improved a lot.

I still recommend making the discussion more concise and improving the flow. i.e. move the limitations towards the end of the discussion rather than early on.

Reviewer #2: Manuscript Title:

Temporal trends in African healthcare system capacities in prevention, detection, response, and sustainability between 2010 and 2023

Summary

This manuscript presents an important and timely analysis of longitudinal trends in WHO SPAR preparedness capacities across 54 African countries from 2010 to 2023. By grouping countries into five UN-defined regions and examining prevention, detection, response, and sustainability domains, the study offers valuable continent-wide insights into preparedness patterns and regional differences.

The topic is highly relevant, particularly in the post-COVID-19 context. The use of a comprehensive, multi-country dataset over more than a decade is a notable strength. However, there are several methodological and analytical issues that should be addressed to strengthen the validity and interpretability of the findings.

Overall, I believe the manuscript has strong potential, but it would benefit from substantial revision.

Major Comments

1. Comparability Across SPAR Reporting Frameworks

The manuscript appropriately notes that SPAR indicators evolved across reporting periods (2010–2017, 2018–2020, and 2021–2023). However, the implications of these structural changes for longitudinal analysis are not fully explored. The observed decline around 2018 may partly reflect revisions in the reporting framework rather than a true decrease in preparedness capacity. Without sensitivity analyses or formal testing for structural breaks, it is difficult to disentangle measurement effects from real trends.

I encourage the authors to consider:

• Conducting sensitivity analyses (e.g., restricting to indicators that remained consistent across periods or focusing on the post-2018 timeframe), and/or

• Performing a structural break analysis to assess the impact of framework revisions.

Clarifying this issue would significantly improve confidence in the reported temporal trends.

2. Analytical Strategy (Regression and Aggregation)

The study uses OLS regression on annual regional means (approximately 14 time points). While this provides a high-level overview, this approach:

• Masks country-level variability,

• Does not explicitly account for the panel nature of the data, and

• Does not address potential serial correlation across time.

Given that country-year data are available, a panel modeling approach (e.g., fixed or random effects) would provide a more robust analytical framework.

If re-analysis is not feasible, I recommend clearly justifying the chosen approach and discussing its limitations more explicitly in the Methods and Discussion sections.

3. Interpretation of COVID-19 Effects

The manuscript suggests that improvements particularly in response capacity after 2018 may reflect the impact of COVID-19. While this interpretation is plausible, it is not formally tested. Without an interrupted time-series or segmented regression design, causal attribution to COVID-19 is not supported by the current analysis.

I suggest either:

• Reframing conclusions to describe temporal associations rather than causal effects, or

• Conducting segmented regression with a clearly defined interruption point (e.g., 2020).

This adjustment would help align the conclusions more closely with the analytical evidence.

4. Self-Reported Data and Reporting Bias

SPAR data are self-reported, and this introduces the possibility of reporting bias particularly in the post-pandemic period when preparedness metrics have gained political visibility. Although the manuscript acknowledges this limitation, the discussion could be expanded to reflect how such bias may influence upward trends in later years.

If possible, triangulating findings with external preparedness indices (e.g., Joint External Evaluation or other global metrics) would strengthen credibility. At minimum, clarifying that the results reflect perceived rather than independently validated preparedness would improve interpretative clarity.

5. Regional Aggregation and Within-Region Variability

Regional averages provide useful summaries but may obscure important within-region heterogeneity. The manuscript does not present measures of dispersion or variability across countries.

Including standard deviations, interquartile ranges, or supplementary variability plots would help readers better understand how evenly (or unevenly) preparedness improvements are distributed within regions.

Minor Comments

• The methodology and discussion sections need to be improved

• The five-point preparedness classification scale appears somewhat arbitrary; please provide justification or consider removing it.

• Clarify the consistency between the stated study period (2010–2023) and analyses emphasizing 2013 onward.

• Consider adjusting for multiple comparisons in the ANOVA analyses.

• Report confidence intervals alongside percent changes where possible.

• To improve reproducibility, consider providing analytical code as supplementary material.

Recommendation

Major Revision

This manuscript addresses an important and policy-relevant topic and has clear potential for publication. However, strengthening the analytical framework and moderating certain interpretations will substantially enhance the robustness and credibility of the findings.

7. PLOS authors have the option to publish the peer review history of their article (what does this mean?). If published, this will include your full peer review and any attached files.

**Do you want your identity to be public for this peer review?** For information about this choice, including consent withdrawal, please see our Privacy Policy.

Reviewer #1: No

Reviewer #2: No

Figure Resubmissions:

---

## [Decision Letter · Decision Letter 2]

27 Apr 2026

Temporal trends in African healthcare system capacities in prevention, detection, response, and sustainability between 2010 and 2023

PGPH-D-25-02770R2

Dear Mr. Sharma,

We are pleased to inform you that your manuscript 'Temporal trends in African healthcare system capacities in prevention, detection, response, and sustainability between 2010 and 2023' has been provisionally accepted for publication in PLOS Global Public Health.

Best regards,

Julia Robinson

Executive Editor

Reviewer Comments (if any, and for reference):

Reviewer's Responses to Questions

**Comments to the Author**

1. If the authors have adequately addressed your comments raised in a previous round of review and you feel that this manuscript is now acceptable for publication, you may indicate that here to bypass the “Comments to the Author” section, enter your conflict of interest statement in the “Confidential to Editor” section, and submit your "Accept" recommendation.

Reviewer #2: All comments have been addressed

2. Does this manuscript meet PLOS Global Public Health’s publication criteria? Is the manuscript technically sound, and do the data support the conclusions? The manuscript must describe methodologically and ethically rigorous research with conclusions that are appropriately drawn based on the data presented.

Reviewer #2: Yes

3. Has the statistical analysis been performed appropriately and rigorously?

Reviewer #2: Yes

4. Have the authors made all data underlying the findings in their manuscript fully available (please refer to the Data Availability Statement at the start of the manuscript PDF file)?

Reviewer #2: Yes

5. Is the manuscript presented in an intelligible fashion and written in standard English?

Reviewer #2: Yes

6. Review Comments to the Author

**Reviewer #2:** The revised manuscript demonstrates substantial improvement and adequately addresses the major methodological concerns raised in the initial review. In particular, the authors have:

• Clarified the implications of SPAR framework changes and incorporated sensitivity analyses (2018–2023) to improve temporal comparability

• Appropriately reframed interpretations of post-pandemic trends to avoid unsupported causal inference

• Justified the use of regional aggregation and clearly delineated the limitations of the analytical approach

• Expanded the discussion of self-reported data bias, including its potential impact on post-pandemic estimates

• Incorporated measures of within-region variability, improving the interpretability of regional summaries

While additional analyses (e.g., structural break testing, panel models, or triangulation with external indices) would further strengthen the work, their absence does not undermine the validity of the current findings. These limitations are now transparently acknowledged and appropriately contextualized.

The manuscript now meets the standards of methodological transparency, analytical coherence, and policy relevance expected for publication. The conclusions are appropriately cautious and aligned with the presented evidence.

I recommend acceptance pending minor editorial refinements.

7. PLOS authors have the option to publish the peer review history of their article (what does this mean?). If published, this will include your full peer review and any attached files.

**Do you want your identity to be public for this peer review?** For information about this choice, including consent withdrawal, please see our Privacy Policy.

Reviewer #2: **Yes:** Chrisogone Justine German
